# JNK-Dependent cJun Phosphorylation Mitigates TGFβ- and EGF-Induced Pre-Malignant Breast Cancer Cell Invasion by Suppressing AP-1-Mediated Transcriptional Responses

**DOI:** 10.3390/cells8121481

**Published:** 2019-11-21

**Authors:** Anders Sundqvist, Oleksandr Voytyuk, Mohamed Hamdi, Herman E. Popeijus, Corina Bijlsma-van der Burgt, Josephine Janssen, John W.M. Martens, Aristidis Moustakas, Carl-Henrik Heldin, Peter ten Dijke, Hans van Dam

**Affiliations:** 1Department of Medical Biochemistry and Microbiology, Science for Life Laboratory, Uppsala University, Box 582, SE-751 23 Uppsala, Sweden; aris.moustakas@imbim.uu.se (A.M.); c-h.heldin@imbim.uu.se (C.-H.H.); P.ten_Dijke@lumc.nl (P.t.D.); 2Department of Cell and Chemical Biology and Oncode Institute, Leiden University Medical Center, P.O. Box 9600, 2300 RC Leiden, The Netherlands; m.hamdi@amsterdamumc.nl (M.H.); h.popeijus@maastrichtuniversity.nl (H.E.P.); cbijlsma77@gmail.com (C.B.-v.d.B.); J.M.Janssen@lumc.nl (J.J.); 3Department of Medical Oncology, Erasmus MC Cancer Institute, Erasmus University Medical Center, 3000 CA Rotterdam, The Netherlands; j.martens@erasmusmc.nl

**Keywords:** invasion, JNK, cJun, TGFβ, AP-1, MAPK, signaling

## Abstract

Transforming growth factor-β (TGFβ) has both tumor-suppressive and tumor-promoting effects in breast cancer. These functions are partly mediated through Smads, intracellular transcriptional effectors of TGFβ. Smads form complexes with other DNA-binding transcription factors to elicit cell-type-dependent responses. Previously, we found that the collagen invasion and migration of pre-malignant breast cancer cells in response to TGFβ and epidermal growth factor (EGF) critically depend on multiple Jun and Fos components of the activator protein (AP)-1 transcription factor complex. Here we report that the same process is negatively regulated by Jun N-terminal kinase (JNK)-dependent cJun phosphorylation. This was demonstrated by analysis of phospho-deficient, phospho-mimicking, and dimer-specific cJun mutants, and experiments employing a mutant version of the phosphatase MKP1 that specifically inhibits JNK. Hyper-phosphorylation of cJun by JNK strongly inhibited its ability to induce several Jun/Fos-regulated genes and to promote migration and invasion. These results show that MEK-AP-1 and JNK-phospho-cJun exhibit distinct pro- and anti-invasive functions, respectively, through differential regulation of Smad- and AP-1-dependent TGFβ target genes. Our findings are of importance for personalized cancer therapy, such as for patients suffering from specific types of breast tumors with activated EGF receptor-Ras or inactivated JNK pathways.

## 1. Introduction

Transforming growth factor-β (TGFβ) family members exert a wide range of biological effects in a variety of cell types, including regulation of proliferation, differentiation, migration, and apoptosis [1,2,3]. Signaling by TGFβ occurs via type I and type II serine/threonine kinase receptors (TGFβRI and TGFβRII, respectively), which mainly propagate the signal through phosphorylation of the receptor-regulated (R-) Smad proteins Smad2 and Smad3 [4,5]. Activated R-Smads form heteromeric complexes with common-partner (Co-) Smad (i.e., Smad4). The complexes accumulate in the nucleus and control gene expression in a cell-type-specific manner through interaction with co-activators and co-repressors and other transcription factors, including members of the AP-1, AP-2, and Ets families [6,7]. Multiple layers of regulation alter both the intensity and duration of the TGFβ response in a cell-type-dependent manner, and also define the specificity of the response [8,9]. Further fine-tuning involves non-Smad signaling pathways such as the ERK1/2, JNK, p38 MAP-kinase and PI3K-AKT-mTOR pathways, which can be induced by TGFβ as well as by other growth-regulatory stimuli [10,11].

In breast cancer, TGFβ has a biphasic role in tumor progression [12,13]. In the early stages, TGFβ/Smad signaling inhibits cell growth and thus acts as a tumor suppressor. In late-stage tumors, TGFβ usually functions as a tumor promoter by stimulating epithelial–mesenchymal transition (EMT)—that is, the de-differentiation of epithelial cells to invasive and metastatic mesenchymal-like cells. These tumor cells may have selectively escaped TGFβ-induced growth inhibitory and apoptotic responses, but have retained or gained certain other responses to TGFβ stimulation. Induction of EMT by TGFβ signaling is promoted by activated Ras, activated Raf, or by serum treatment [14], and involves the TGFβ-Smad-inducible transcriptional repressors Snail and Slug [15,16,17]. EMT can promote the development of cancer cells that express stem-cell markers and exhibit stem cell characteristics, and therefore may metastasize through acquired invasiveness and enhanced self-renewal potential [18]. Moreover, upregulation of TGFβ stimulates a favorable microenvironment for rapid tumor growth [1,19,20]. Pro-oncogenic effects of TGFβ are also mediated via non-Smad signaling pathways that are initiated directly by TGFβR activation or indirectly by Smad-driven transcriptional responses that stimulate the expression of growth factors like TGF-α/epidermal growth factor (EGF) and platelet-derived growth factor (PDGF) [21].

The dimeric Jun/Fos and Jun/ATF AP-1 transcription factor complexes are composed of cJun, JunB, JunD, c-Fos, FosB, Fra1, Fra2, and certain ATF members (e.g., ATF2, ATFa/7, and ATF3). These proteins control cell proliferation, differentiation, and survival by regulating gene expression in response to a large number of stimuli and pathways (e.g., the ERK, JNK, and p38 MAP-kinase pathways) [22,23,24,25,26,27,28]. Certain AP-1 family members have been implicated in tumor cell invasion [24,29,30], and in particular Fra1 has been associated with breast cancer metastasis, EMT, and cancer stemness [31,32,33,34]. The dimer composition and activity state of the AP-1 complexes in both tumor cells and various stromal cell types appear critical, but the mechanisms are still largely unknown [25,26,35,36]. 

Using a 3D model of collagen-embedded spheroids of non-malignant, pre-malignant, and metastatic MCF10A human breast cancer cells, we found that TGFβRI kinase, Smad3, Smad4, and the AP-1 components cJun, JunB, cFos, and Fra1 co-operate in TGFβ-induced invasion in the presence of EGF; this co-operation induced the expression of various genes, including *MMP10, SERPINE1*, *WNT7A*, and *WNT7B* [37,38]. Non-malignant and pre-malignant MCF10A cells require EGF to proliferate [39,40] and to efficiently migrate and invade in the presence of TGFβ and oncogenic Ras. At least part of the EGF requirement is related to the strong EGFR-mediated activation of the MEK-ERK1/2 pathway. The latter increases the levels of members of the AP-1 family, in particular of the Fos subfamily. In the absence of EGFR-MEK signaling, TGFβ/Smad activation cannot induce critical Smad-AP-1-dependent EMT and invasion-associated genes [41]. In addition to the MEK-ERK1/2 MAP-kinase pathway, both EGF and TGFβ can also activate the MKK-JNK MAP-kinase pathway, and thereby phosphorylate and activate the AP-1 component cJun. However, in contrast to the EGFR-MEK-ERK1/2 pathway, the MKK4-JNK-cJun pathway can be defective in human cancers due to loss-of-function MKK4 mutations [42,43,44,45,46], and JNK negatively affects breast tumorigenesis and mammary cell motility and migration [47]. 

In the present study, we examined the role of JNK-dependent cJun phosphorylation in the pro-oncogenic TGFβ response in the premalignant MCF10A-RAS (MII) breast cancer model. Studies with phospho-deficient, phospho-mimicking, and dimer-specific cJun mutants showed that the N-terminal phosphorylation of cJun by JNK strongly inhibits its ability to induce migration and invasion, and to activate multiple AP-1- and Smad-dependent TGFβ target genes. These results show that MEK, JNK and phospho-cJun exhibit distinct pro- and anti-invasive functions in breast cancer cells through differential regulation of TGFβ- and EGF-induced invasion/migration genes.

## 2. Materials and Methods 

### 2.1. Cell Culture

MCF10A MII cells were obtained from Dr. Fred Miller (Barbara Ann Karmanos Cancer Institute, Detroit, MI, USA) and maintained at 37 °C and 5% CO_2_ in DMEM/F12 (Gibco, Thermo Fisher Scientific, Stockholm, Sweden), supplemented with 5% fetal bovine serum (FBS) (Biowest, Almeco A/S, Esbjerg, Denmark), 20 ng/mL epidermal growth factor (EGF) (PeproTech, EC Ltd, London, UK), 100 ng/mL cholera toxin (Sigma-Aldrich AB, Stockholm, Sweden), 0.5 μg/mL hydrocortisone (Sigma-Aldrich AB, Stockholm, Sweden), and 10 μg/mL insulin (Sigma-Aldrich AB, Stockholm, Sweden). F9 cells [48,49] were cultured in F12-DMEM (1:1) containing 9% fetal calf serum (FCS), penicillin, streptomycin, and 0.1 mM β-mercaptoethanol. HeLa cells [50,51] were grown in Dulbecco’s modified Eagle’s medium (DMEM) containing 9% FCS, 100 μg/mL penicillin, and 100 μg/mL streptomycin.

### 2.2. Reagents and Antibodies

Recombinant human TGFβ1 and EGF were from PeproTech. The following kinase inhibitors were used at the indicated concentrations: the TGFβ type I kinase inhibitors SB505124 (2.5 μM; Sigma-Aldrich AB, Stockholm, Sweden) and LY394946 (2.0 μM; Calbiochem-Merck, Stockholm, Sweden), the MEK1 inhibitors PD184352 (0.5 μM; Sigma-Aldrich AB, Stockholm, Sweden) and AZD6244 (0.25 μM; Selleckchem, Houston TX 77230 USA ), and the JNK1/2 inhibitors SP600125 (10 μM; Calbiochem) and JNK-IN-8 (2.5 μM; Selleckchem, Houston TX 77230 USA). Puromycin was purchased from Invitrogen and used at a concentration of 0.5 µg/mL. 

Antibodies against the following epitopes were used: phospho-Tyr1068 EGFR (#3777; Cell Signaling Technology, Leiden, the Netherlands), phospho-Thr202/Tyr204 Erk1/2 (#4370 Cell Signaling Technology, Leiden, the Netherlands), FN1 (F3648; Sigma-Aldrich AB, Stockholm, Sweden), cFos (sc-52; Santa Cruz Biotechnology, Inc., Santa Cruz, CA, USA), Fra1 (sc-22794; Santa Cruz, CA, USA), HA (sc-805; Santa Cruz), phospho JNK (#4668; Cell Signaling Technology), cJun (sc-1694; Santa Cruz, CA, USA), phospho-Ser63 cJun (#9261; Cell Signaling Technology, Leiden, the Netherlands), MKP1 (sc-1102; Santa Cruz, CA, USA), phospho-Ser178 paxillin (#A300-100A; Bethyl Laboratories, Montgomery, TX 77356 USA), PAI1 (#612024; BD Transduction Laboratories, San Jose, CA, USA,), Smad2/3 (#610843, BD Transduction Laboratories, San Jose, CA, USA,), and phospho-Ser423/425 Smad3 (#9520; Cell Signaling Technology, Leiden, the Netherlands).

### 2.3. 3D Spheroid Invasion Assays

Spheroid invasion into collagen was performed as described previously [37,38,52]. Briefly, all spheroids consisted of 10^3^ cells. Single spheroids were embedded in a 1:1 mix of neutralized collagen and complete medium supplemented with 12 mg/mL of methylcellulose, and allowed to polymerize on the top of neutralized collagen in a 96-well-plate. TGFβ1 was directly added to the embedding solution. Pictures were taken at day 0, day 1, and day 2 after embedding and quantified by measuring the area occupied by cells using Adobe Photoshop CS3 software (Adobe Photoshop CS3 Extended, version 10.0, Adobe, San Jose, CA, US, 2011).

### 2.4. In Vitro Wound-Healing Assay 

For the wound healing assay, 3 × 10^5^ MCF10A MII cells per well were seeded in a 6-well plate. Media were changed to starvation media (complete media, supplemented by 0.2% FBS) after which cells were grown until ~90% confluence. The cell monolayer was wounded by crossed scratching using a 200 μL pipette tip. After washing with phosphate-buffered saline (PBS), the cells were incubated with the indicated agents, and the same crossed scratch wounds were photographed at the indicated time points using an inverted-phase-contrast microscope (Zeiss Axiovert 40CFL, Carl Zeiss Microimaging GmbH, Jena, Germany). TScratch software (version 1.0, CSElab, Zurich, Switzerland) was used for quantification of the scratch wound; eight measurements per sample were performed.

### 2.5. Immunoblot Analysis

For the immunoblot analysis, 5 × 10^5^ cells were seeded in 6-well-plates. The following day, cells were incubated for 16–24 h in EGF-containing serum-starvation medium (EGF, insulin, cholera toxin, hydrocortisone, 0.2% FBS) or in starvation medium lacking EGF, as indicated, and stimulated with 5 ng/mL of TGFβ1 (PeproTech, EC Ltd, London, UK) for the indicated time periods. Cells were lysed in 2 × SDS loading buffer and subjected to SDS-PAGE and immuno-blotting, as described previously [37].

### 2.6. RNA Isolation, cDNA Synthesis, and Quantitative Real-Time-PCR

Total RNA was isolated by RNeasy Kit (QIAGEN AB, Sollentuna, Sweden). cDNA was prepared using the iScript kit (Bio-Rad Laboratories AB, Solna, Sweden) using 0.5 µg of total RNA, according the manufacturer’s instructions. The cDNA samples were diluted 10 times with water. Two microliters of cDNA was used in 12 µL quantitative real time-PCR reactions with appropriate primers and KAPA SYBR FAST qPCR kit Master Mix (PCR Biosystems, London, UK) and BioRad CFX96 real-time PCR detection system according the manufacturer’s instructions. All samples were analyzed in triplicate for each primer set. Gene expression levels were determined with the comparative ΔΔCt method and the non-stimulated condition was set to 1. Relative expression levels were normalized to *GAPDH*, and presented as mean ± SD. The complete list of primers can be found in Appendix A.

### 2.7. DNA Transfer and Constructs 

F9 cells were transfected by the calcium phosphate method and HeLa cells by the DEAE-dextrane method [50,51]. Luciferase activity was determined according to the manufacturer’s protocol (Promega Corp., Stockholm, Sweden). As controls for transfection efficiency and proper expression of effector proteins, aliquots of the same lysate from which luciferase activity was determined were analyzed by immuno-blotting. Lentiviruses were produced as described previously [53]. MCF10A MII cells were plated on 6-cm dishes 24 h prior to infection. Cells were infected at 50–70% confluency by removing growth media from the cells and adding to each dish a mixture of 1.5 mL of virus supernatant, 1.5 mL of complete media, and hexadimethrine bromide (Polybrene, Sigma-Aldrich AB, Stockholm, Sweden) at a final concentration of 8 µg/mL.

The luciferase reporter constructs -517/+63 *MMP1* (collagenase) Luc and 5xGal4-tata-Luc have been described [54]. To create the -1977/-1858-urokinase-tata pGL3 reporter, the *Xho*I/*Sma*I- fragment of the pBL-tata-CAT 5 construct was replaced by the *Xho*I/(blunted) BamHI fragment of pGL3 basic (Promega Corp., Stockholm, Sweden). The -1977/-1858-*PLAU* (urokinase)-tata was subsequently constructed by replacing the *Bam*HI-*Kpn*I fragment of -1977/-1858-uPA-TK-CAT4 [55] with the *Bam*HI-*Kpn*I fragment of tata-pGL3.

The pCMV-cJun-HA expression vector and the corresponding Ala and Asp mutants (wt, 7A, and 6D) were kindly provided by D. Bohmann and M. Musti [56,57,58]. HA-tagged versions were constructed by replacing the *Hpa*I-*Kpn*I fragments with the corresponding fragment of pCMV-cJun-HA. The cJun(TAD)GCN4(DBD) construct (j4) was obtained from S. Oliviero [59]. The leucine zipper mutant encoding plasmids pCMV-cJun-m1-J/A-zip-HA, -m1,5-J/A-zip-HA, and -m2,5-J/F-zip-HA were created by replacing the *Pst*I-*Kpn*I fragments of pCMV-cJun-HA (wt TAD), cJun-7A, and cJun-6D with the corresponding fragments of pRSV-c-Jun-m1-HA, -m1,5-HA, and -m2,5-HA [60]. The distinct cJun-HA cDNAs were subsequently cloned into the PGK/IRES-neo cassette of the lentiviral vector pLVbc-neo [61]. pMT2, pMT2-cdc42^V12^, and pMT2-ΔMEKK were kindly provided by A. Hall.

### 2.8. Statistical Analysis

Collagen invasion assays contained n ≥ 6 spheroids for each condition, and were repeated at least twice with similar results. Migration assays were repeated at least three times with similar results. Data are presented as means ± SD. The differences between experimental groups were analyzed using Welch’s *t*-test, with **p* < 0.05, ***p* < 0.01, and ****p* < 0.001 being considered significant. For immunoblots and qRT-PCR, at least three independent experiments were performed. For luciferase activity and qRT-PCR, the differences between experimental groups were analyzed using Welch’s *t*-test, with **p* < 0.05, ***p* < 0.01, and ****p* < 0.001 being considered significant.

## 3. Results

### 3.1. JNK-Dependent Phosphorylation of cJun Negatively Affected MCF10A MII Cell Migration 

Previously, knockdown experiments showed that in the presence of EGF, TGFβ induces invasion of MCF10A MII cells by critically inducing the expression of cJun and JunB [37]. To examine the role of JNK-dependent cJun phosphorylation, we compared the activities of wild-type (wt) cJun with the phosphorylation defective cJun-7A and phospho-mimicking cJun-6D mutants, in which nearly all the established and potential JNK phosphorylation sites are replaced by either alanine or aspartic acid residues (Figure 1a). The phospho-mimicking Asp-substituted cJun-6D has been shown to exhibit potent gain-of-function properties, for instance in neuronal apoptosis, whereas cJun-7A showed loss-of-function behavior in these assays [56,57,58,62]. Stable lentiviral overexpression of wt cJun, cJun-6D, and cJun-7A mutants in MCF10A MII cells cultured in the presence of EGF (Figure 1b) showed that cJun-6D did not enhance migration in cell culture wound healing assays, and in some assays even inhibited migration induced by EGF or by TGFβ together with EGF (Figure 1c; Appendix A). In contrast, cJun-7A potently enhanced both EGF- and TGFβ-induced migration (Figure 1c; Appendix A). Interestingly, wt cJun behaved in these assays with intermediate impact, which might reflect variations in its phosphorylation state between experiments. Importantly, the cJun-7A cells did not show enhanced proliferation under these conditions (Appendix A). These results suggested that JNK-dependent cJun phosphorylation can inhibit cJun-dependent migration. To validate this possibility, we inhibited endogenous cJun phosphorylation in MII cells with a lentiviral vector stably expressing the JNK-specific variant of the MAPK phosphatase MKP1 (also called dual-specificity phosphatase 1—DUSP1) [53,63]. MKP1 is a nuclear protein that can efficiently de-phosphorylate and thereby inhibit nuclear JNK, but is unlikely to inhibit JNK in the cytoplasm and at the cell membrane.

Indeed, as shown in Figure 2a,c, MCF10A MII cells expressing JNK-specific MKP1 showed severely reduced levels of Ser63-phosphorylated cJun, but not of phosphorylated paxillin (PXN), an established JNK substrate localized in cytoplasm and plasma membrane [64]. Ectopic expression of this MKP1 mutant also showed enhanced migration compared to the control cells (Figure 2b).

To examine whether the stimulatory effect of JNK-specific MKP1 on cell migration was dependent on N-terminal cJun phosphorylation, we co-infected cells with MKP-1 and cJun-6D- expressing viruses (Figure 2c), as the phospho-mimicking Asp residues of cJun-6D cannot be affected by MKP1 and JNK. Importantly, cJun-6D blocked the enhancing effect of MKP1, both in the presence and absence of TGFβ (Figure 2d). Together, these data show that the JNK-dependent phosphorylation of cJun had an inhibitory effect on EGF- and TGFβ-dependent migration of MCF10A MII cells in wound healing assays.

### 3.2. EGF and TGFβ Signaling Had Different Effects on JNK-Dependent cJun Phosphorylation

We next examined the effects of EGF and TGFβ on cJun phosphorylation. As shown in the time-course analysis in Figure 3a and the schematic summary in Figure 3d, after 30 min stimulation only EGF induced strong phosphorylation of cJun-Ser63, which was detected in several upshifted gel electrophoresis retarded bands, indicating additional JNK-dependent phosphorylations of other residues [57,58] (see below). In contrast, both EGF and TGFβ induced the levels of total cJun, and the ratio of the levels of P-Ser63-cJun and (total) cJun was much lower at the later time points (Figure 3a). cFos was also efficiently induced when EGF was added. In line with this, only EGF was found to induce the levels of phosphorylated active JNK and ERK, whereas efficient induction of Smad3 phosphorylation and plasminogen activator inhibitor 1 (PAI1) expression only occurred after TGFβ treatment (Figure 3a).

Next, we compared the effects of TGFβ stimulation on the levels of P-S63-cJun and (total) cJun in the absence and continuous presence of EGF. As shown in Figure 3b, both P-S63-cJun and cJun were increased by TGFβ under these conditions, but in line with the results shown in Figure 3a, the relative ratio of the levels of P-Ser63-cJun and (total) cJun was strongly reduced at the late time points (12 h and 24 h).

Analysis of specific kinase inhibitors confirmed the effects of TGFβRI and JNK on cJun-Ser63 phosphorylation. In MCF10A MII cells stimulated with TGFβ in the presence of EGF, the phosphorylation of cJun-Ser63 was hardly affected by two different TGFβRI kinase inhibitors, despite a reduction in total cJun, whereas it was completely inhibited by the two JNK inhibitors used (Figure 3c). Interestingly, inhibition of MEK enhanced cJun-Ser63 phosphorylation, in line with other studies in breast cancer cells [46].

### 3.3. N-Terminal Phosphorylation of cJun Negatively Affected the Activation of MMP1 and MMP10

Our previous knockdown and overexpression experiments in MCF10A MII cells showed that induction of cJun by TGFβ and EGF is critical for activation of a specific subset of TGFβ/Smad-induced EMT and invasion genes, including *MMP1* and *MMP10*, whereas *MMP2* was suppressed by cJun [37]. To examine the role of JNK-dependent cJun phosphorylation on the expression of these *MMP* genes, we first compared the effects of wt cJun, non-phosphorylatable mutant cJun-7A, and phospho-mimicking mutant cJun-6D. Transiently overexpressed cJun wt and cJun-6D activated *MMP1* and *MMP10* only moderately; however, cJun 7A enhanced *MMP1* and *MMP10* much more efficiently (Figure 4a). The effects of these mutants on *MMP1* and *MMP10* expression thus reflected their effects on cell migration, as shown in Figure 1c. No differences were seen for *MMP2* and *SERPINE1,* encoding PAI1 (Figure 4a).

The apparent negative effect of cJun phosphorylation on *MMP1* and *MMP10* was intriguing, since work of various groups has shown that the phosphorylation of Thr91/93 and/or Ser63/73 by JNK rather enhances the transactivation function of cJun, in addition to regulating its stability [26,65]. Therefore, we next examined wt cJun as well as cJun-7A and -6D mutants for their abilities to activate an *MMP1*-promoter-driven luciferase reporter plasmid in the presence and absence of ΔMEKK, a constitutively active inducer of JNK that lacks its N-terminal inhibitory domain. We performed this experiment in mouse embryonal F9 cells, which do not express endogenous cJun, and in which the N-terminal JNK target sites of ectopically expressed cJun are only very weakly phosphorylated in the absence of ΔMEKK or other strong JNK inducers, such as ultraviolet (UV) light (Figure 4b) [49,66]. Indeed, in the absence of ΔMEKK, both wt cJun and cJun-7A could efficiently activate the *MMP1* promoter, but in the presence of this inducer of N-terminal cJun phosphorylation the activation by wt cJun was reduced by about 75%, whereas activation by the phosphorylation-defective cJun-7A was not affected (Figure 4c). The phospho-mimicking mutant cJun-6D activated the *MMP1* reporter only weakly, which was not further inhibited by ΔMEKK. These experiments thus indicate that hyper-phosphorylation of the cJun N-terminus by JNK suppresses both EGF- and TGFβ+EGF-dependent migration (Figure 2b,d), as well as the ability of cJun to efficiently activate specific Smad- and AP-1-dependent genes (Figure 4a,c).

### 3.4. JNK-Dependent Phosphorylation of cJun Specifically Inhibited Gene Activation and Migration by Jun/Fos Dimers

As mentioned above, previous studies concluded that the phosphorylation of multiple residues in the cJun N-terminus, including Ser63 and Ser73, is associated with the activation, rather than inhibition, of cJun-inducible promoters [66,67,68]. One example is the *JUN* promoter, which is activated by heterodimers of cJun and ATF2-related factors, and maybe by cJun homodimers [50,51,69,70,71]. Moreover, homodimeric cJun hybrid proteins in which the cJun transactivation domain (TAD) is fused to the DNA-binding domain (DBD) of Gal4 or of growth hormone factor (GFH)1 activate Gal4- and GHF1-dependent promoters more strongly after phosphorylation of Ser63 and Ser73 [49,66,72]. However, *MMP1* and *MMP10*—the genes we found to be repressed by JNK-dependent cJun-phosphorylation—are activated by Jun/Fos and/or Jun/Fra heterodimers, also upon TGFβ-Smad stimulation [6,37]. This raised the possibility that the effects of N-terminal phosphorylation on cJun-dependent transactivation are dimer-specific. To examine this possibility, we analyzed the role of the cJun DNA-binding and dimerization domains in the inhibition of cJun upon JNK-dependent phosphorylation. 

First, we examined a hybrid AP-1 transcription factor in which the cJun DBD was replaced by the DBD of the yeast AP-1 homologue GCN4, which can efficiently bind to AP-1 binding sites as a homodimer [59]. Importantly, activation of the *MMP1* promoter in F9 cells by this cJun-GCN4 fusion protein was not inhibited by ΔMEKK (Figure 4d), supporting the notion that the inhibitory effect of N-terminal cJun phosphorylation on cJun-dependent transactivation is dimer-specific. Next, we analyzed one previously published and two as-yet unpublished cJun leucine zipper mutants that preferentially dimerize with either cFos- or ATF2-like proteins (Figure 5a) [60,73]. In mammalian one/two-hybrid assays in cervical cancer HeLa cells transfected with a Gal4-dependent reporter, the basic region-leucine zipper (bZIP) DBDs of cJun-m1 and c-Jun-m1,5 fused to the Gal4 DBD were found to activate the reporter efficiently only in the presence of ATF2-VP16, but not with cFos (Figure 5b,c). In contrast, the bZIP DBD of cJun-m2,5 could not be activated by ATF2-VP16, and showed the opposite effect (Figure 5b,c). We subsequently examined full-length versions of these cJun dimerization mutants for their effects on the -1977/-1858 *urokinase*/*PLAU* enhancer, which contains both a cJun/ATF2 and a cJun/Fos site [55]. To investigate the effect of JNK-dependent cJun phosphorylation, this analysis was done both in the absence and presence of cdc42^V12^—a constitutively active upstream activator of the JNK pathway which strongly enhances N-terminal cJun phosphorylation in these HeLa cells (Figure 5d,f). Importantly, transactivation by wt cJun and the cJun-m2,5 mutant—which can both dimerize with Fos—was strongly inhibited upon activation of the JNK pathway by cdc42^V12^, whereas transactivation by the ATF2-preferring mutants cJun-m1 and cJun- m1,5 was much less affected (Figure 5e,f). Moreover, the observed inhibition in the presence of cdc42^V12^ was completely dependent on the presence of the JNK phosphorylation sites in cJun, as the phospho-site-deficient cJun 7A mutants were not inhibited (Figure 5e).

Thus, these data strongly suggest that the N-terminal phosphorylation of cJun by JNK specifically represses transactivation by cJun molecules forming heterodimers with Fos or Fra proteins (Figure 5f). Comparison of the transactivating potential of phospho-deficient (7A) and phospho-mimicking (6D) versions of the Fos- and ATF2-preferring cJun-m2,5 and cJun-m1 mutants (Figure 5g) on the urokinase/*PLAU* reporter further supported the conclusion that the inhibitory effect of JNK-dependent phosphorylation is cJun/Fos-dimer specific (Figure 5h). We next compared the dimer-specific phospho-mimicking 6D and non-phosphorylatable 7A cJun mutants for their effects on breast cancer cell migration. Upon lentiviral infection in MCF10A MII cells, both the ATF-preferring mutants (i.e., cJun-m1-6D and cJun-m1-7A) inhibited TGFβ-induced migration (Figure 5i). In contrast, only the 6D variant of the Fos-preferring mutant cJun-m2,5 inhibited invasion. Moreover, similar to what was observed for the 6D and 7A mutants with a wildtype leucine zipper (Figure 1c), cJun-m2,5-7A strongly enhanced basal and TGFβ-induced migration, whereas m2,5-6D had no effect (Appendix A). The cells expressing cJun-m2,5-7A and 6D also strongly differed when assayed for their abilities to induce cells in spheroids embedded in collagen to invade (Figure 6a). cJun-m2,5-7A induced basal invasion, whereas cJun-m2,5-6D inhibited TGFβ-induced invasion. This difference was reflected on the molecular level; the faster migrating and invading cJun-m2,5-7A cells contained decreased levels of endogenous Ser63-phosphorylated cJun, and increased levels of phosphorylated active ERK1/2 and paxillin. In contrast, the TGFβ-induced levels of fibronectin (FN) were more enhanced in the slower migrating and invading Jun-m2,5-6D expressing cells (Figure 6b).

To obtain more information about the mechanism by which JNK-dependent cJun phosphorylation can influence cell migration and invasion, we have previously used chromatin immunoprecipitation (ChIP)-seq and RNA-seq and specific inhibitors to analyze the expression of various Smad and AP-1 target genes previously found to be induced in TGFβ- and EGF-stimulated cells, including several *MMPs*, *WNT7*, and genes encoding EGF family ligands [37,38,52,75]. Without TGFβ stimulation, *MMP1, HBEGF*, and *MMP10* showed higher expression in MII cells transfected with cJun-m2,5-7A than in cJun-m2,5-6D transfected cells, whereas *WNT7A* and *MMP2* were expressed at comparable levels and *WNT7B* was expressed at lower levels (Figure 7a,b). Upon TGFβ treatment *MMP1, MMP10*, and *WNT7B* showed higher expression in the cJun-m2,5-7A transfected cells than in the cJun-m2,5-6D transfected cells (Figure 7a,b). These differences in the expression of *WNT7* and EGF family ligands might explain the different levels of phospho-ERK1/2 in cells transfected with cJun-7A and -6D (Figure 6b). In fact, we also observed corresponding differences in the levels of cFos and phosphorylated EGFR (Figure 7c). Finally, analysis of 166 estrogen receptor (ER) positive human breast cancer cases showed enhanced levels of *WNT7B* in breast cancers with loss-of-function mutations in the upstream JNK pathway components MAP2K4 and MAP3K1 (Figure 7d).

In conclusion, these results show that the hyper-phosphorylation of cJun by JNK inhibits a subset of TGFβ-induced migration and invasion genes by inhibiting cJun/Fos-dependent gene activation.

## 4. Discussion

TGFβ signaling is known to exhibit dual functions in tumor progression, acting as a tumor suppressor in early stages, while functioning as a tumor promoter in later stages. However, the molecular mechanisms by which these functions are exhibited, and how they depend on the genetic context and tumor microenvironment, are less clear. Important in this respect is that TGFβ signaling is mediated via both Smad and non-Smad signaling pathways, and that many of the non-Smad pathways—including the different MAPK pathways and their AP-1 transcription factor targets—can also be strongly activated by other growth factors and cytokines, such as EGF and interleukin 1 (IL1) [22,23,24,25]. 

Inhibitors of TGFβ that are currently in preclinical development for treatment of metastatic cancer inhibit all TGFβ activities. Although clinical efficacy has been observed, systemic application may result in on-target side effects, such as cardiovascular toxicity [76,77]. Therefore, more specific intervention in aberrant TGFβ signaling in diseases is needed. Inhibitors of the non-Smad signaling MAP-kinase pathways ERK1/2, JNK, and p38 as well as PI3K-AKT-mTOR are also being tested for cancer treatment. However, these pathways have by themselves profound effects on proliferative, apoptotic, and differentiation pathways, and deregulation can contribute to chemotherapeutic drug resistance, proliferation of cancer-initiating cells (CICs), and premature aging [65,78,79]. For instance, JNK and its substrate cJun can exert pro- and anti-oncogenic functions in different cell types and stages of cancer development [25,35,47,64,65,80,81]. It is therefore essential to mechanistically understand the functions of MAP-kinase family members in different tumor types, including their interplay with TGFβ-Smad signaling. Important in this respect is also that in basal-like breast cancers, the PI3K-AKT and Ras-Raf-MEK-ERK1/2 pathways are often activated, whereas luminal A breast cancers do not show activation of the ERK1/2 pathway, but often contain loss-of-function mutations in the JNK pathway [42,43,44,45].

Here, we show that the expression of various pro-invasive TGFβ and EGF-Ras-MEK-ERK1/2 target genes, in particular the Jun/Fos dependent genes *MMP1* and *MMP10*, can be enhanced upon JNK inhibition or overexpression of a non-phosphorylatable cJun mutant. This shows that JNK signaling can exhibit negative effects on various genes involved in migration and invasion. Interestingly, we previously found that *MMP1*, *MMP10*, and other pro-invasive TGFβ-induced genes are late Smad-AP-1 target genes that are only efficiently induced by TGFβ in the presence of EGF [37,41]. In line with this, efficient and prolonged TGFβ induction of cJun protein and, in particular, Fos family members also required EGF (Figure 3a,b; [41]). However, combined TGFβ and EGF treatment also induced much higher levels of phospho-cJun at the early time points (Figure 3a,b), which can counteract the pro-invasive effects. Prolonged or additional JNK activation at later time points might thus suppress critical pro-invasive genes. Via this mechanism, JNK might inhibit TGFβ-induced EMT and invasion in some cell types, similar to what is seen in mouse trophoblasts, where JNK inhibition was found to induce *snai2* and *mmp* genes and to promote EMT [82]. Moreover, although JNK signaling is required for normal mouse mammary gland development, JNK1 and JNK2 can act as suppressors of mammary tumor development and JNK1 + JNK2 double knock-out cells from these mice show enhanced motility and migration [47].

In addition to *MMP1* and *MMP10*, we found *HBEGF, WNT7A*, *WNT7B*, and *FLNA* to be regulated differently by phosphorylated cJun, as compared to non-phosphorylated cJun. These genes might also be negatively regulated by JNK to suppress cell migration and invasion in certain cell types. Moreover, we recently identified both *WNT7A* and *WNT7B* as JunB- and Smad4-dependent TGFβ target genes that can stimulate the TGFβ-induced collagen invasion of MII cells [38]. 

In line with the specific suppression of Jun/Fos target genes by JNK, dimer-specific mutants of cJun that do not bind to Fos family proteins were not inhibited by JNK-dependent phosphorylation or phospho-mimicking cJun mutations. Notably, the ability of these mutants to bind ATF is in agreement with the well-established positive effect of JNK on cJun/ATF2 [65,83,84]. Interestingly, a recent study showed that JNK can suppress tumor formation via ATF2-dependent gene expression and that ATF2-dependent gene expression is frequently downregulated in human cancers [81]. Moreover, ATF2 might directly or indirectly effect cJun/Fos-dependent gene expression and thereby inhibit Jun/Fos-driven pro-oncogenic events [60,74]. However, ATF2 does not seem to suppress the TGFβ-induced collagen invasion of MCF10A MII cells [37].

It remains to be established how the hyperphosphorylation of cJun by JNK specifically inhibits Jun/Fos and not Jun/ATF transcriptional activity. Upon stable lentiviral infection with cJun-m2,5-6D, the levels of phospho-ERK1/2 and cFos were decreased, but this may have been an indirect effect. The interaction between cJun and cFos is mediated through their leucine zipper domains and, as far as known, is not affected by N-terminal cJun phosphorylation. However, N-terminal hyper-phosphorylation of cJun might decrease the affinity of cJun/Fos for Fos-family specific co-activators and chromatin-remodeling factors, such as Trip6 [85] and BAF proteins [27], or increase the affinity for cJun/Fos-specific co-repressors [71,72]. Alternatively, in chromatin-bound multi-protein complexes that require the presence of both Jun/Fos dimers and Fos-interacting transcription factors, cJun hyper-phosphorylation may cause destabilization and reduce the interaction of these complexes with chromatin and/or chromatin remodelers [27,28].

Together with our previous study on TGFβ–EGFR cooperation, this study shows that MEK and JNK exhibit distinct pro- and anti-invasive functions through the differential regulation of Jun, Fos, and ATF components of AP-1 and, thereby, of Smad- and AP-1-dependent TGFβ target genes. The anti-invasive effects of JNK may contribute to the tumor-suppressive properties of JNK. This is important for personalized cancer therapy for patients with breast tumors having MEK-ERK1/2 and MKK-JNK pathway mutations. Breast cancers commonly retain or gain TGFβ-Smad tumor promoting properties, indicating that they promote tumor progression [13]. Loss-of-function mutations in the JNK pathway found in specific tumor subsets may therefore counteract the negative effect of high JNK activity and phospho-cJun on TGFβ-Smad-induced tumor cell migration/invasion and other tumor-promoting properties. In line with this, we observed enhanced levels of *WNT7B* in breast cancer cells with loss-of-function mutations in MAP2K4 and MAP3K1. This further stresses the notion that clinical inhibitors of TGFβ and JNK may either inhibit or enhance tumor progression, depending on other oncogenic defects and genetic background. In this respect, it is interesting to mention that in hepatocellular carcinoma JNK has been found to promote an inflammatory microenvironment supporting tumorigenesis, but to inhibit carcinogenesis in the hepatocytes themselves [80]. Both JNK and cJun have been found to positively or negatively regulate genotoxic-stress-induced cell cycle arrest and apoptosis, depending on the agent, dose, and cellular context [25,65,83,84]. Our results may therefore be important for the reported inhibitory effects of TGFβ on the tumor response to target-specific and genotoxic anti-cancer drugs. The potentially cell-type-specific mechanisms of TGFβ-induced drug resistance remain to be elucidated, but EMT, MEK-ERK1/2, MKK-JNK, and AP-1 seem to be involved [34,46,86,87,88,89,90].

In summary, we identified specific anti-oncogenic functions of the JNK pathway in the Smad-AP-1-regulated EMT and invasion of pre-malignant breast cancer cells, and have identified various target genes involved in tumor promotion. These observations may be important for future personalized cancer therapeutic strategies.

## Figures and Tables

**Figure 1 cells-08-01481-f001:**
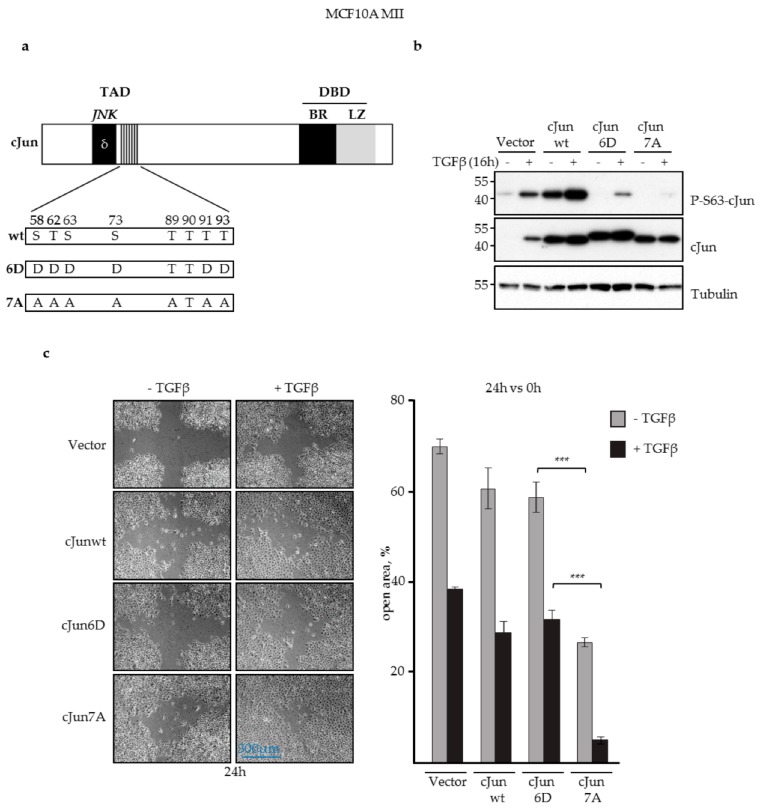
JNK-dependent phosphorylation of cJun negatively affected MCF10A MII cell migration. (**a**) Schematic representation of cJun showing the known and potential N-terminal JNK phosphorylation sites and the changes in the 7A loss-of-function and 6D gain-of function mutants. TAD, transactivation domain; DBD, DNA-binding domain; δ, JNK docking site; BR, basic region; LZ, leucine zipper. Numbers correspond to serine/threonine phosphorylation sites in the TAD; (**b**) Immunoblot analysis of P-S63-cJun and cJun upon stable lentiviral overexpression of cJun wild-type (wt), cJun-6D, and cJun-7A in MCF10A MII cells cultured in the presence of EGF and treated with 5 ng/mL TGFβ for 16 h, as indicated. Tubulin was analyzed as loading control; (**c**) Migration (% open area) of the MCF10A MII cells stably overexpressing wt cJun as well as cJun-6D and cJun-7A mutants, in the presence or absence of TGFβ (5 ng/mL) for 24 h, as measured by wound healing assays; ****p <* 0.001.

**Figure 2 cells-08-01481-f002:**
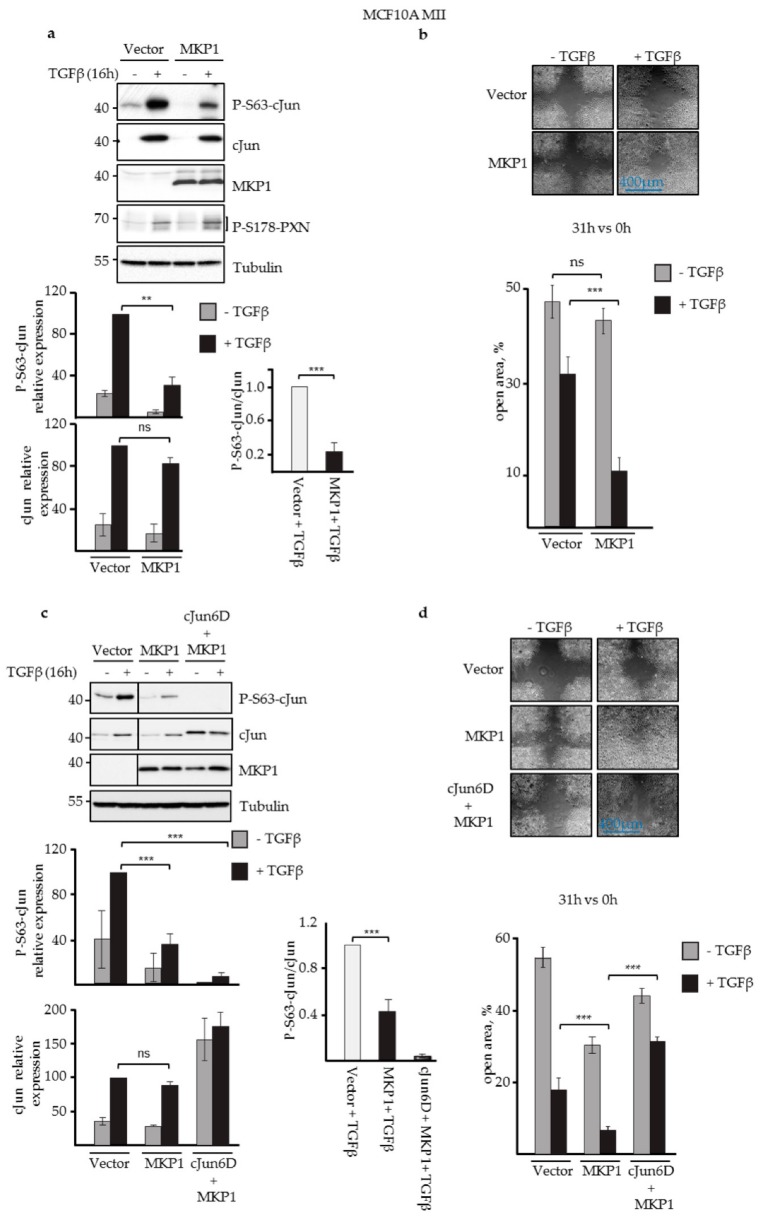
JNK-specific MKP1 inhibited cJun phosphorylation and enhanced MCF10A MII cell migration. (**a**) Immunoblot analysis of P-S63-cJun, cJun, MKP1, and P-S178-paxillin (PXN) levels upon stable lentiviral overexpression of JNK-specific MKP1 in MCF10A MII cells and treatment with 5 ng/mL TGFβ for 16 h, as indicated. Tubulin was included as loading control; a representative experiment is shown. The average levels of P-S63-cJun and (total) cJun (quantified by densitometry and normalized to loading control) of five independent biological replicates are depicted in the graphs. For proper comparison, the TGFβ-induced P-S63-cJun and cJun levels obtained for the vector control of each experiment were set at 100. The effect of JNK-specific MKP1 on the relative ratio of the TGFβ-induced levels of P-S63-cJun and cJun is also shown, ***p* < 0.01, ****p* < 0.001. (**b**) Migration (% open area) of the MCF10A MII cells stably overexpressing JNK-specific MKP1, as measured by wound healing assays in the absence or presence of TGFβ for 31 h. (**c**) Immunoblot analysis of MCF10A MII cells stably overexpressing cJun-6D and/or JNK-specific MKP1, in the absence and presence of TGFβ. A representative experiment is shown; the line indicates where the blot was cut; all samples were run on one gel. The average levels of P-S63-cJun, cJun, or cJun-6D (quantified by densitometry and normalized to loading control) of three independent biological replicates are depicted in graphs. For proper comparison, the TGFβ-induced P-S63-cJun and cJun levels obtained for the vector control of each experiment were set at 100. The effect of JNK-specific MKP1 on the relative ratio of the TGFβ-induced levels of P-S63-cJun and cJun or cJun-6D is also shown. (**d**) Migration of the MCF10A MII cells stably overexpressing cJun 6D and/or JNK-specific MKP1, as measured by wound healing assays in the absence or presence of TGFβ for 31 h.

**Figure 3 cells-08-01481-f003:**
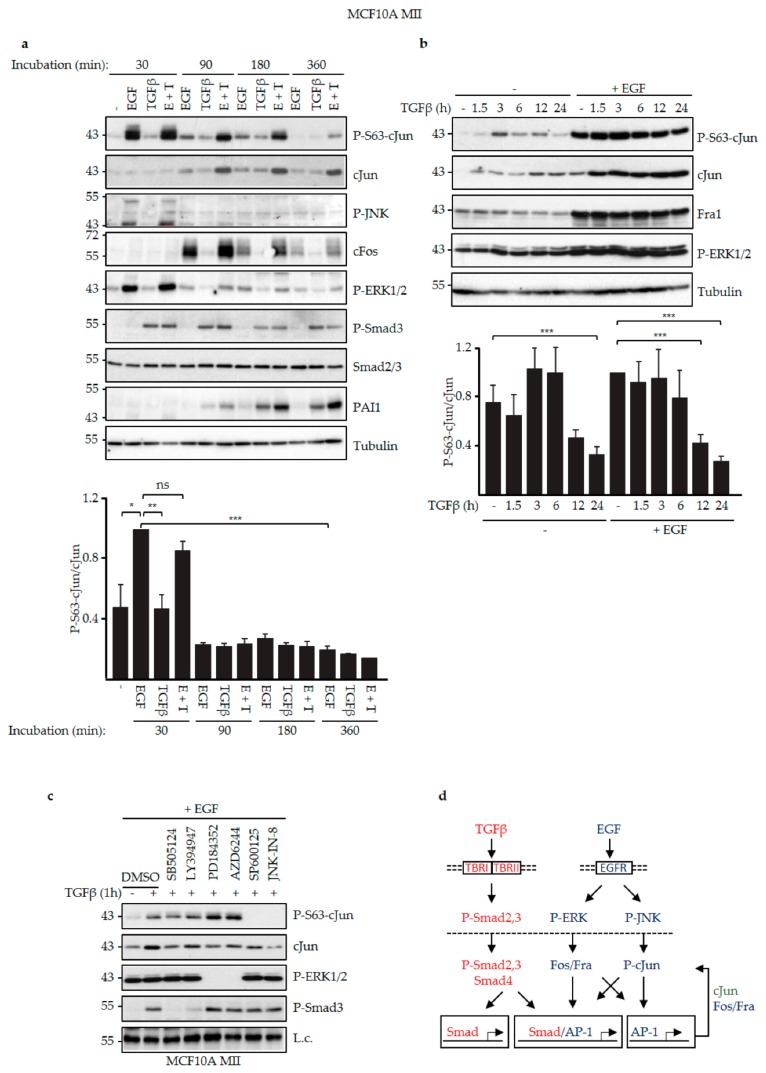
EGF and TGFβ differentially induced phospho-cJun activity. (**a**) MCF10A MII cells were incubated for 24 h in serum-starvation medium (insulin, cholera toxin, hydrocortisone, 0.2% FBS) without EGF, subsequently treated with 20 ng/mL EGF and/or 5 ng/mL TGFβ for the indicated time periods, and then analyzed by immunoblotting for P-S63-cJun, cJun, P-JNK, cFos, P-ERK1/2, P-Smad3, Smad2/3, and PAI1. Tubulin was included as loading control; a representative experiment is shown. The levels of P-S63-cJun and (total) cJun of three independent biological replicates were quantified by densitometry to determine the effects of EGF and TGFβ on the relative ratio of the levels of P-S63-cJun and cJun in time, depicted in the graph. For proper comparison of the data of the different experiments, the ratio of the levels of P-S63-cJun and cJun at the 30 min EGF time point of each experiment was set at 1.0. (**b**) MCF10A MII cells were incubated for 24 h in EGF-containing serum-starvation medium (EGF, insulin, cholera toxin, hydrocortisone, 0.2% FBS) or in starvation medium lacking EGF as indicated, treated with 5 ng/mL TGFβ for the indicated time periods, and analyzed by immunoblotting for P-S63-cJun, cJun, Fra1, and P-ERK1/2. Tubulin was included as loading control; a representative experiment is shown. The levels of P-S63-cJun and (total) cJun of four independent biological replicates were quantified to determine the effects of EGF and/or TGFβ on the relative ratio of the levels of P-S63-cJun and cJun in time, depicted in the graph. For proper comparison of the data of the different experiments, the ratio of the levels of P-S63-cJun and cJun at the 0 h TGFβ + EGF time point of each experiment was set at 1.0. **p* < 0.05, ***p* < 0.01, ****p* < 0.001 (**c**) MCF10A MII cells were incubated for 24 h in EGF-containing serum-starvation medium, subsequently stimulated for 1 h with TGFβ in the presence or absence of the indicated TGFβRI, MEK, and JNK inhibitors, and analyzed by immunoblotting for P-S63-cJun, cJun, P-ERK1/2, and P-Smad3. The inhibitors were added 30 min before TGFβ. A background band was included as loading control (L.c.); (**d**) Schematic model to explain the data in panels (a–c) in view of the current literature. EGF signaling via cell-membrane-localized EGFR triggers phosphorylation and activation of ERK and JNK, which in the nucleus activate and induce FOS family members and phosphorylation of cJun, respectively, which subsequently can auto-regulate their own expression via AP-1 sites [22,26]. TGFβ signaling via membrane-localized TBRI and TBRII triggers phosphorylation and activation of Smad2 and 3, and thereby can both directly and indirectly induce TGFβ target genes controlled by Smad and/or AP-1 sites [1,2].

**Figure 4 cells-08-01481-f004:**
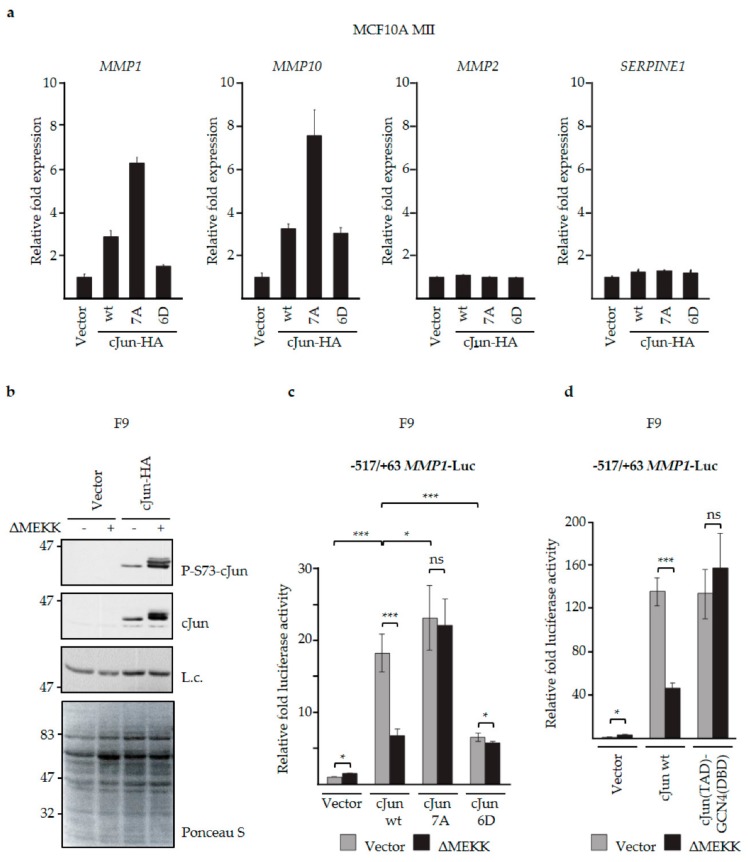
N-terminal cJun phosphorylation negatively affected the activation of *MMP1* and *MMP10*. (**a**) MCF10A MII cells were transfected with expression vectors for wt cJun as well as cJun-6D and -7A mutants, and *MMP1, MMP10, MMP2,* and *SERPINE1* mRNA levels were analyzed by qPCR and normalized to *GAPDH*, which was not significantly affected by the treatments; (**b**) F9 cells were transfected with expression vectors for constitutively active ΔMEKK and cJun-HA as indicated. After 6 h, cell extracts were prepared for immunoblotting analysis with antibodies specific for phosphorylated cJun-Ser73 or total cJun. A background band of the cJun blot (L.c.) and Ponceau S staining were included as loading control; (**c**) F9 cells were transfected with the -517/+63 pGL3 *MMP1* luciferase reporter and expression vectors for constitutively active ΔMEKK and wt cJun, and cJun-6D or -7A mutants, as indicated. After 6 h, cell extracts were prepared and luciferase activity was analyzed; **p* < 0.05, ****p* < 0.001. (**d**) F9 cells were transfected with the -517/+63 pGL3 *MMP1* luciferase reporter and expression vectors for constitutively active ΔMEKK and cJun wt or cJun-GCN4—a chimeric protein in which the DNA-binding domain (DBD) of cJun is replaced by the corresponding DNA-binding domain of GCN4, as indicated (TAD: transactivation domain). After 16 h, cell extracts were prepared and luciferase activity was analyzed.

**Figure 5 cells-08-01481-f005:**
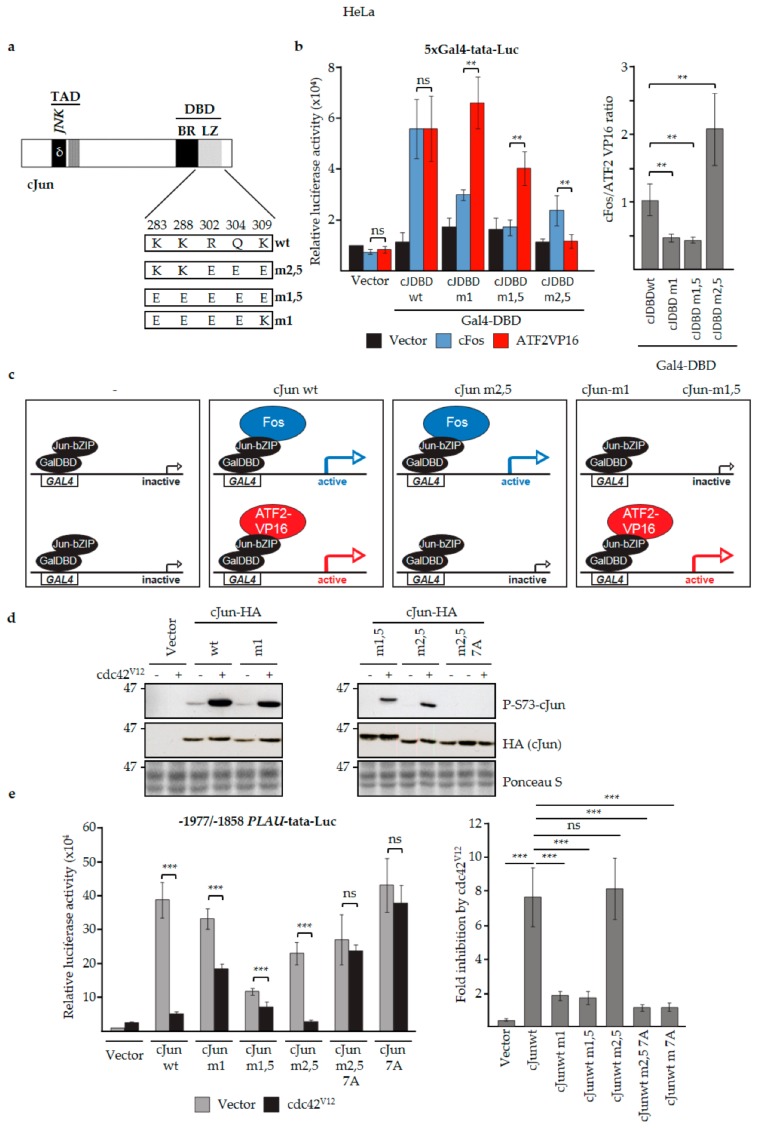
JNK-dependent hyperphosphorylation of cJun specifically inhibits gene activation and migration by Jun/Fos dimers. (**a**) cJun mutants with altered Jun/Fos and Jun/ATF2 dimerization specificities. Schematic representation showing the changes in the mutants m2,5, m1,5, and m1 compared to wt cJun. Numbers correspond to the positively charged cJun amino acid residues at the e and g positions of the cJun leucine zipper (LZ) α-helix that stabilize cJun/Fos heterodimers [60,74], some of which were mutated to negatively charged amino acid residues. (**b**) Left: HeLa thymidine kinase (tk)^−^ cells were cotransfected with the 5xGal4-tata luciferase reporter, the indicated wt and mutant GAL4DBD-cJunDBD (cJDBD) fusion constructs and/or ATF2-VP16 or cFos vectors, and luciferase activity was measured. Right: the luciferase activity ratio of the fold activation by Fos and ATF2VP16 on the indicated Jun constructs. cJDBD, the basic region-leucine zipper (bZIP) DNA-binding domains of the cJun mutants m2,5, m1,5, and m1 or wt cJun fused to the Gal4 DBD, as depicted schematically in panel (**c**); **p* < 0.05, ***p* < 0.01, ****p* < 0.001. (**c**) Schematic representation of the cJun DBD mammalian-one-hybrid analysis [60] described in panel (**b**). GAL4 represents one of the five GAL4 transcription factor binding sites of the 5xGal4-tata luciferase reporter; GalDBD-JunbZIP represents the transcriptionally inactive wt and mutant GAL4DBD-cJunDBD (cJDBD) fusion constructs used in (b) which bind to the GAL4 site but by themselves do not activate (left panels). The upper panels show the activation of the reporter by GalDBDJunbZIP-wt and -m2,5 when cFos is co-expressed; the lower panels show the activation by GalDBDJunbZIP-wt, -m1, and -m1,5 when ATF2VP16 is co-expressed. ATF2VP16 is used because ATF2 wt is not active in these assays. (**d**) HeLa tk^-^ cells were transfected with expression vectors for cdc42^V12^ (a constitutively active upstream activator of the JNK pathway) and wt and mutant cJun-HA, as indicated. After 16 h, cell extracts were prepared for immunoblot analysis with antibodies specific for phosphorylated cJun-Ser73 or cJun-HA. (**e**) Left: HeLa tk^-^ cells were transfected with the -1977/-1858 *PLAU* tata luciferase reporter, cdc42^V12^, and the (HA-tagged) cJun vectors, as indicated. After 16 h, cell lysates were prepared. Right: the fold inhibition by cdc42^V12^ on the different cJun constructs; average of at least six independent biological replicates. (**f**) Schematic representation of the regulation of the -1977/-1858-*PLAU* enhancer by cJun/Fos and cJun/ATF2 dimers in the absence and presence of cdc42^V12^, a constitutively active upstream activator of the JNK pathway. The upper panels depict the observed enhancer activation by cJun/Fos (formed by cJun wt and cJun m,2,5) and cJun/ATF (formed by cJun wt, cJun m1, and cJun m,1,5) in the absence of cdc42^V12^. The lower panels show the observed enhancer activation upon phosphorylation of JNK and cJun in response to cdc42^V12.^. Under these conditions, enhancer activation by cJun/Fos (formed by cJun wt and cJun m,2,5) was severely reduced. (**g**) Schematic representation of the combined phospho-site (Figure 1a) and dimerization (Figure 5a) mutants of cJun used. (**h**) HeLa tk^-^ cells were transfected with the -1977/-1858 *PLAU* tata luciferase reporter and HA-tagged wt cJun, and the cJun-7A and -6D mutants, as indicated. After 16 h cell lysates were prepared. (**i,j**) Migration of MCF10A MII cells stably overexpressing the HA-tagged Fos- (m2,5) or ATF- (m1) preferring variants of cJun-6D and cJun-7A, as measured by wound-healing assays for 37 h (**i**) and immunoblot validation of (HA) expression levels (**j**).

**Figure 6 cells-08-01481-f006:**
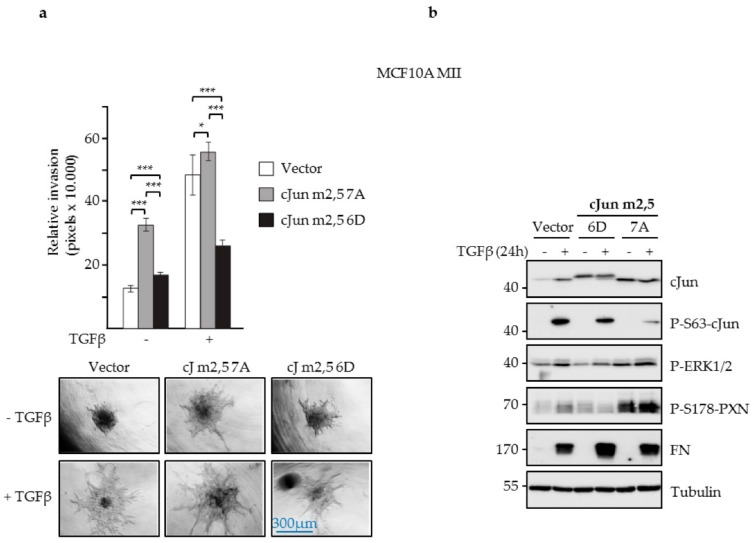
Effects of cJun dimerization-specific mutants, and phosphorylation-deficient- and mimicking mutants on collagen invasion. (**a**) Collagen invasion (pixels x 10.000) of MCF10A MII spheroids stably overexpressing cJun-(cJ)-m2,5-6D or cJun-m2,5-7A. Pictures of spheroids were taken 32. h after embedding (bottom) and relative invasion was quantified as the mean area that the spheroids occupied (top); **p* < 0.05, ****p* < 0.001. (**b**) Immunoblot analysis of the MCF10A MII cells overexpressing cJun-m2,5-6D or cJun-m2,5-7A.

**Figure 7 cells-08-01481-f007:**
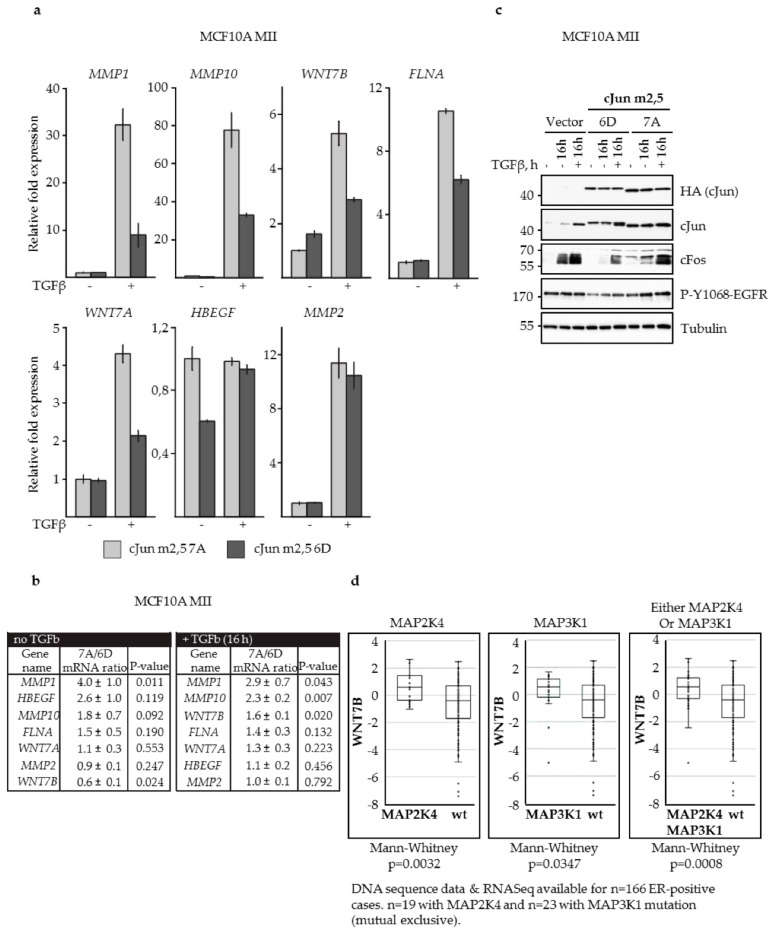
Effects of cJun dimerization-specific, and phosphorylation-deficient and -mimicking mutants on the expression of TGFβ -Smad and AP-1 target genes. (**a**,**b**) Differential effects of cJun-m2,5-6D or cJun-m2,5-7A on TGFβ-inducible gene expression. MCF10A MII cells stably overexpressing cJun-m2,5-6D or cJun-m2,5-7A were incubated in EGF-containing starvation medium for 16 h and treated, or not, with TGFβ (5 ng/mL) for 16 h, after which mRNA levels were analyzed by qPCR and normalized to *GAPDH*. In panel (b) the ratio of the expression in cJun-m2,5-7A cells versus cJun-m2,5-6D cells is depicted. (**c**) Immunoblot analysis of MCF10A MII cells overexpressing cJun-m2,5-6D or cJun-m2,5-7A, treated with TGFβ (5 ng/mL) or mock-treated, as indicated. (**d**) Analysis of 166 estrogen-receptor-positive (ER+) human breast cancer cases showing enhanced levels of *wnt7b* in breast cancers cells with loss-of-function mutations in the upstream JNK pathway components MAP2K4 and MAP3K1.

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
