# Peer review of "JNK-Dependent cJun Phosphorylation Mitigates TGFβ- and EGF-Induced Pre-Malignant Breast Cancer Cell Invasion by Suppressing AP-1-Mediated Transcriptional Responses"

_cells, 2019, doi:10.3390/cells8121481_

Round 1
Reviewer 1 Report
Brief Summary: In previous studies, Sundqvist et al have shown that in TGFb-stimulated feed-forward network pathway, which induces the aggressiveness in breast cancer, is governed by the cooperation between JUNB and SMAD2/3. Based on the previous finding, Sundqvist et al again showed that JNK-dependent phosphorylation of cJun represses a subset of TGFb -related aggressive genes and results in the inhibition of cell migration and invasion. The inhibition effect is cJun/Fos-dimer specific. The study provided a possible explanation of pro- and anti-oncogenic effect of TGFb, may through the crosstalk between TGFband EGFR which regulates differential regulation of downstream genes.
Comments:
Overall, the finding in this paper is very interesting and also includes comprehensive description. Few comments could be considered in this paper:
In Figure 1b, cells were treated with TGFb and the indicated proteins were analyzed by immunoblot. However, in the article (line 200) is written that cells were cultured in the presence of EGF. This discrepancy needs to be clarified.The authors claimed that the negative effect on migration is dimer-specific, how would the phosphorylation of cJun on N-terminal inhibits Jun/Fos transcription activity specifically? There seems a missing link between phosphorylation of cJun and dimer-specificity.
How does the co-stimulation of EGF and TGFb results in differential regulation and have distinct pro- or anti-invasive functions? The time frame of JNK activation and prolong activation could be critical.
Figure 5c, 5f and 7d are not readable.
Reviewer 2 Report
The manuscript submitted for evaluation undertakes an important subject - the role of JNK-dependent cJun phosphorylation in the pro-oncogenic TGFβ response in breast cancer. The experiments are well planned and the text well written.
However, the authors did not avoid several mistakes
1. Authors should perform WB for all replicates of presented experiments and show they in the graphs (mean value with SD).
2. The description of adding TGFb should be standardized in all graphs and figures. TGFb - and TGFb + would be better.
3. Pictures in Fig 2b are illegible and seem not to correspond to the presented graph. Authors should present more representative microphotographs. The resolution should be corresponded to presented in Fig 1.
4. The representative migration pictures were omitted in Fig 2d. I would like to ask authors for adding that microphotographs.
5. Fig 5c and 5f - diagrams are not legible. Authors should enlarge them or improve their resolution.
5. Fig 5j - I would like to ask the authors about more representative control blot. The currently presented is too long exposed.
Reviewer 3 Report
The manuscript by Sundqvist et al aim to demonstrae that MEK-AP-1 and JNK-phospho-cJun have distinct pro- and anti-invasive functions, and that they act through differential regulation of Smad- and AP-1-regulated by TGFβ. Despite the potential interest of the work, there are many experimental problems that need to be addressed:
Migration assay presented in figure 1 is hardly understandable, apart from the fact that the pictures are of low quality, is difficult to understand how was possible to clearly measure the migration rate. HSP90 is widely used as loading control, I don't think it's correct given that this protein has been involved/related to several pathophysiology mechanisms in breast cells. In figure 4B, the author are using a "background" band as loading control, I do not believe this is a correct way to verify the quality of the loading.Minor point
1. Figure 1 is split between 2 pages, please adjust it.
Round 2
Reviewer 2 Report
The authors improved only the resolution of the drawings, while their substantive quality is still very low and difficult to interpret.
The method of presenting WB results in their current form disqualifies these results
Reviewer 3 Report
Response 1. We used a Tscratch software to calculate the percentage ratio of open area between time “0” and corresponding time point. We have added in the Material and Methods section about the origin of the software (lines 138 and 139). This software automates the analysis of wound healing assays performed with a range of cell lines with differing cell morphology (Gebäck T. et al, Biotechniques (2009),46(4):265-74) and it has been widely used for investigation of wound healing assay image data sets.
Despite this clarification, I'm not convince that the cells are actually migrating, by observing the image, I can detect more a proliferation than migration. Software are design to give you a results, but they can't be apply without previous consideration of the experimental settings.
Response 2. In many of our experiments, we observed that Hsp90 showed the same protein levels as other commonly used loading controls, such as Ponceau S staining and α-Tubulin. We used Hsp90 because it was convenient to use it since its molecular weight is quite different from the molecular weight of the proteins that we were analyzing. As an example, we have added Hsp90 and α-Tubulin loading control for comparison in the updated Fig. 5j.
As the authors now have added proper loading control you should use a-tubulin in the manuscript.
Regarding the other points I'm satisfied with author's reply
